# Crosstalk between ILC3s and Microbiota: Implications for Colon Cancer Development and Treatment with Immune Check Point Inhibitors

**DOI:** 10.3390/cancers15112893

**Published:** 2023-05-24

**Authors:** Fabiana Drommi, Alessia Calabrò, Grazia Vento, Gaetana Pezzino, Riccardo Cavaliere, Fausto Omero, Paola Muscolino, Barbara Granata, Federica D’Anna, Nicola Silvestris, Claudia De Pasquale, Guido Ferlazzo, Stefania Campana

**Affiliations:** 1Laboratory of Immunology and Biotherapy, Department Human Pathology “G.Barresi”, University of Messina, 98122 Messina, Italyriccardo.cavaliere@unime.it (R.C.); claudia.depasquale@unime.it (C.D.P.); scampana@unime.it (S.C.); 2Department of Experimental Medicine (DIMES), University of Genoa, 16132 Genova, Italy; grazia.vento@edu.unige.it; 3Unit of Experimental Pathology and Immunology, IRCCS Ospedale Policlinico San Martino, 16132 Genova, Italy; 4Medical Oncology Unit, Department of Human Pathology “G.Barresi”, University of Messina, 98125 Messina, Italynicola.silvestris@unime.it (N.S.)

**Keywords:** type 3 innate lymphoid cells, microbiota, colon cancer, immunotherapy, immune check point inhibitor, IL-22, tertiary lymphoid structures

## Abstract

**Simple Summary:**

In recent years, growing evidence has suggested that the gut microbiome can significantly influence antitumor immunity, both within and outside the gastrointestinal tract, thereby affecting the efficacy of immune checkpoint inhibitors (ICIs. A link between microbiota composition and response to ICIs has been reported in both mouse and human studies. Gut microbial features depend on a delicate balance of tolerance for commensal microbiota and defense against various potentially pathogenic microbiota orchestrated by host immune system. Type 3 innate lymphoid cells (ILC3s) are a group of tissue-resident innate lymphocytes related to host immune cell–microbiome interactions. They orchestrate immunity, inflammation and tolerance in the intestines and any alterations in their functions can cause gut inflammation, colon cancer and immunotherapy resistance.

**Abstract:**

Type 3 innate lymphoid cells (ILC3s) are primarily tissue-resident cells strategically localized at the intestinal barrier that exhibit the fast-acting responsiveness of classic innate immune cells. Populations of these lymphocytes depend on the transcription factor RAR-related orphan receptor and play a key role in maintaining intestinal homeostasis, keeping host–microbial mutualism in check. Current evidence has indicated a bidirectional relationship between microbiota and ILC3s. While ILC3 function and maintenance in the gut are influenced by commensal microbiota, ILC3s themselves can control immune responses to intestinal microbiota by providing host defense against extracellular bacteria, helping to maintain a diverse microbiota and inducing immune tolerance for commensal bacteria. Thus, ILC3s have been linked to host–microbiota interactions and the loss of their normal activity promotes dysbiosis, chronic inflammation and colon cancer. Furthermore, recent evidence has suggested that a healthy dialog between ILC3s and gut microbes is necessary to support antitumor immunity and response to immune checkpoint inhibitor (ICI) therapy. In this review, we summarize the functional interactions occurring between microbiota and ILC3s in homeostasis, providing an overview of the molecular mechanisms orchestrating these interactions. We focus on how alterations in this interplay promote gut inflammation, colorectal cancer and resistance to therapies with immune check point inhibitors.

## 1. Introduction

The human gastrointestinal (GI) tract harbors a vast community of microorganisms, the gut microbiota, which exerts a marked influence on host immunity in both homeostasis and disease [1]. Indeed, the microbiota and immune system engage in extensive bidirectional communication. Signals derived from the gut microbiota are critical for the development of both the innate and adaptive immune system, while the immune system helps to keep host–microbial mutualism in check [2]. A proficient way to study the interactions between the microbiota and immune system is to use germ-free (GF) models. Studies in GF mice have demonstrated that the absence of the gut microbiota is associated with the profound impairment of gastrointestinal-associated lymphoid tissue (GALT) development and reduced numbers of intra-epithelial lymphocytes (IELs) [3], as well as reduced levels of IgA antibodies [4]. On the other hand, the immune system is able to recognize conserved microbial patterns and integrate these signals to promote strong tolerance mechanisms. Growing evidence has revealed the significant contribution of RAR-related orphan receptor (RORγt)-expressing cells in a variety of tolerance mechanisms in the intestinal microbiota [5]. A distinct population of lymphocytes in the gut, termed type 3 innate lymphoid cells (ILC3s), is composed of at least two different subsets, both relying on RORγt and Id2 for their development: lymphoid-tissue inducer (LTI) cells and NCR+ ILC3. In mice, LTi cells can be characterized by the expression of the chemokine receptor CCR6 [6] and a subpopulation expressing CD4 represents the most mature LTi cells. On the other hand, CCR6 ILC3s express NKp46 and depend on the master regulator T-bet [7]. However, ILC3s are highly plastic and additional transitional states have been observed. Among these states, CCR6–NKp46 ILC3s have been demonstrated to differentiate into NKp46+ ILC3s. In humans, however, it is still unclear how to distinguish LTi cells from NCR+ ILC3s. All human RORγt+ ILC3s can express CCR6 [8] and can be classified into different subsets based on the expression of CD56 and NKp44 [8,9]. According to this classification, LTi/LTi-like cells are most likely represented by the CD56neg ILC3 fraction [8]. Similarly, IL-22 production is also somewhat different between the human and mouse ILC3 subsets. While both ILC3 subsets in mice are capable of producing IL-22, IL-22 production in humans is mainly confined to NKp44+ populations [9].

LTI cells are crucial for the development of lymphoid structures through the release of lymphotoxin (LT) molecules and interactions with mesenchymal stromal cells (MSCs), while IL-22-producing ILC3 populations express the natural cytotoxicity receptor NKp44 [10] and are able to reinforce intestinal epithelial integrity. ILC3s are mostly found on mucosal surfaces and help to maintain appropriate immune responses to the intestinal microbiota by regulating intestinal epithelial cells and both innate and adaptive immune cell populations [5]. Thus, it can be envisaged that the dysregulation of ILC3s resulting from environmental perturbations goes along with impaired host–microbial mutualism and can lead to aberrant immune activation and associated intestinal inflammation [11]. On the other hand, intestinal inflammation due to alterations in the composition and function of the microbiota (dysbiosis) is believed to impact immune surveillance for colorectal cancer (CRC) and its progression, as well as the response to novel immunotherapies [12]. In this review, we provide an overview of the crosstalk occurring between ILC3s and microbiota and discuss their potential implications in CRC development and response to immune checkpoint inhibitor (ICI) therapy.

## 2. Crosstalk between ILC3s and Microbiota

### 2.1. Effects of the Microbiota on ILC3s

Commensal microbes have a crucial impact on ILC3 behavior, both through direct stimulation and, indirectly, the regulation of epithelial and myeloid cell responses [13]. Signals derived from commensal microbes can be directly sensed by specific immune cell receptors, such as toll-like receptors (TLRs). Spits and co-workers detected TLR transcript encoding for types 1, 2, 5, 6, 7 and 9 (but not 3 or 4) in human ILC3s [14]. However, they reported that only TLR2 was functional and that its stimulation could induce IL-2 that acted in an autocrine manner to increase IL-22 expression by ILC3s [14]. Besides TLRs, RORγt^+^ ILCs can directly recognize commensal bacteria through the engagement of natural cytotoxicity receptors (NCRs), such as NKp44, NKp46 and NKp30, which have been recently identified as another pathway for the direct recognition of PAMPs (pathogen-associated molecular patterns) [15,16]. Among NCRs, NKp44 triggering induces TNFα release by ILC3s, whereas the combined engagement of NKp44 with cytokine stimulations are able to induce IL-22 secretion [17,18].

Gut microbiota-derived tryptophan metabolites, such as indole and indole-3-acid-acetic (IAA), are ligands for the aryl hydrocarbon receptor (AhR), a ligand-dependent transcription factor expressed by RORγt+ ILCs. AhR activation is essential for the development and function of intestinal RORγt+ group 3 ILCs [19]. AhR activation promotes IL-22 expression through cooperative interactions with RORγt, which directly binds at the IL-22 promoter to induce its transcription [20]. AhR-mediated IL-22 production by ILC3s promotes protective responses following DNA damage in epithelial stem cells and the postnatal development of isolated lymphoid follicles and cryptopatches. In another study [21], an AhR agonist, in combination with IL-1β, stimulated ILC3s to produce IL-22, whereas an AhR antagonist caused the differentiation of ILC3s into IFNγ-expressing ex-ILC3s, confirming the conserved function of AhR in supporting ILC3 maintenance. Additionally, AhR activation supports the expansion of ILC3s promoting the transcription of Notch 1/2 and the expression of anti-apoptotic proteins, such as Bcl-2 and Ki67 [22]. In Ahr^−/−^ mice, both the percentage and absolute number of RORγt^+^ ILCs are significantly decreased, indicating that AhR deficiency could lead to the defective expansion of ILC3s [19]. Metabolites generated by the fermentation of dietary components by the microbiota could be ligands for specific receptors expressed on the surface of ILC3s. Among these, short-chain fatty acids (SCFAs), such as acetate (C2), propionate (C3) and butyrate (C4), are known to influence ILC3 functions since they express G protein-coupled receptors (GPCRs). ILC3s are able to sense SCFAs via Ffar2 (GPR43), Ffar3 (GPR41) and GPR109a [23]. Ffar2 mainly regulates CCR6+ ILC3 function, which is a highly proliferative ILC3 subset in colonic cryptopatches, isolated lymphoid follicles and larger colonic patches [24]. In particular, acetate mainly promotes ILC3 proliferation via Ffar2 through the activation of the AKT, STAT3 and mTOR signaling pathways, whereas propionate preferentially stimulates IL-22 production via Ffar3. IL-22 production is also indirectly supported by acetate via the upregulation of IL-1β receptor expression on ILC3s. In line with these findings, it has been reported that the genetic ablation of Ffar2 (Ffar2^ΔRorc^) significantly impairs the proliferation and production of IL-22 by CCR6^+^ ILC3s, resulting in hindered host defense and reduced tissue repair [24]. Butyrate promotes IL-22 release by RORγt^+^ ILCs through the GPR41 pathway [25] and can inhibit IL-17-producing ILC3s by interacting with GPR109a, which is expressed by dendritic cells (DCs) [26]. In addition, microbiota-derived butyrate is a regionally specialized factor suppressing ILC3s in terminal ileal Peyer’s patches (PPs). Indeed, GPR109a is expressed highly in RORγt+ PP ILC3s and butyrate treatment significantly reduces the amount of IL-22 produced by ILC3 PP cells [27]. Additionally, GPR109a signaling negatively regulates ILC3 expansion. In mice treated with niacin, a Gpr109a agonist, the frequency of ILC3s is reduced, whereas the deletion of this receptor is associated with higher numbers of ILC3s [25]. Another report showed that the *Helicobacter Species* functions as a negative regulator of ILC3s in the large intestine. In particular, *H. typhlonius* and *H. apodemus* promote the loss of colonic T-bet- and RORγt-expressing ILC3s, thereby inducing gut dysbiosis [28].

The microbiota can also influence ILC3 activity through the modulation of myeloid and epithelial cell responses in the GI tract [29]. Indeed, the microbiota triggers IL-1β production through the Myd88- and Nod2-dependent pathways via intestinal CX3CR1+ macrophages. This cytokine can induce the production of GM-CSF from ILC3s or IL-23 via CD103+ CD11b+ DCs, which increases IL-22 production by ILC3s [30]. Upon microbial stimulation, CX3CR1+ macrophages can also release the TNF-like ligand 1A (TL1A), whose binding with death receptor 3 (DR3) expressed on human ILC3s promotes not only their proliferation but also their IL-22 production when combined with cytokine stimulation [31,32]. Overall, the microbiota is able to sustain IL-22 production by ILC3s through various mechanisms; however, on the other hand, the same microbiota tightly regulates this cytokine by constantly limiting its release to prevent the deleterious effect of IL-22. Mechanistically, the microbiota can induce IL-25 release by epithelial cells, which reduces the amount of IL-22 produced by ILC3s in a DC-dependent manner. The inhibitory mechanism mediated by IL-17RB+ (IL-25R) on DCs requires physical interactions with ILC3s and is independent from IL-23 [5].

Interestingly, commensal bacteria can also directly affect the antigen presenting capability of ILC3s. In particular, under steady-state conditions, the microbiota can induce IL-23 production by mononuclear phagocytes (DCs and macrophages), a crucial signal for the reversible silencing of MHCII+ in ILC3s, thereby reducing the capacity of ILC3s to present antigens to T cells in the intestinal mucosa [33].

Thus, the gut microbiota can significantly affect ILC3 functions through various mechanisms (Figure 1), but further studies are still required to more specifically understand the microbial signals that activate, suppress or fine-tune ILC3s in the gut.

### 2.2. Effects of ILC3s on the Microbiota

ILC3s play a pivotal role in preserving the homeostasis of the gut microbiota by regulating physical, biochemical and immunologic barriers. In the intestinal tract, ILC3s are a dominant source of IL-22 and interact with intestinal epithelial cells (IECs) to modulate responses to the microbiota and promote bacteria anatomical containment [34]. In this regard, Sonnenberg et al. [11] demonstrated that Rag1^−/−^ mice that were treated with anti-IL-22 mAb showed the dissemination of *Alcaligens* (commensal bacteria) from the intestinal lymph nodes to the spleen and liver, causing systemic inflammation and a loss of intestinal homeostasis. Another study [35] showed that in IL 22^−/−^ mice, tight-junction proteins (claudine and occludine), mucins and antimicrobial peptides (S100A8 and S100A9, β-defensin, RegIIIβ, RegIIIγ) were reduced in epithelial cells, thus resulting in the loss of the physical separation of the microbiota from the intestinal epithelium [36]. Moreover, IL-22 produced by ILC3s promotes intestinal epithelial fucosylation by inducing the expression of fucosyltransferase 2 (Fut2). This process offers survival advantages to beneficial bacteria that use fucose for their colonization and as an energy source. Indeed, in RORγt IL-22-deficient mice, Fut2 expression is significantly reduced and the mice become more susceptible to infection [37]. Besides IL-22, ILC3s can also induce epithelial fucosylation via lymphotoxin α (LTα) production and, accordingly, the numbers of F-ECs and Fut2 expression are reduced in Lta^−/−^ mice. A positive feedback loop exists between membrane-bound LTβ (LTα1β2) and IL-22 since LTβ produced by ILC3s triggers IL-23 secretion by DCs, in turn inducing IL-22 production by ILC3s [38]. So far, only a few studies have investigated gut microbial species in deficient mice or altered ILC3 populations. However, higher levels of *Segmented filamentous bacteria* (SFB), as well as *Clostridiales* species, have been reported in the absence of ILC3s (RORcCre × Id2fl/fl) [21]. IL-22 and LTα produced by ILC3s can also influence microbiota composition [37,39]. In the absence of Ahr, outgrowths of SFB have been observed due to the loss of IL-22 produced by ILC3s, indicating the role of this lymphocyte population in the regulation of microbiota composition. Along the same line, in both Lta^−/−^ and IL-22^−/−^ mice, the expansion of SFB and a concomitant decrease in *Bacteroides* have been found [40], whereas in another study, IL-22 depletion resulted in a marked decrease in commensal *Lactobacilli* [35].

ILC3s can also interact with adaptive immune cells to modulate the quality and magnitude of immune responses to commensal microbiota. Recent studies have highlighted the critical role of ILC3s in negatively selecting microbiota-specific CD4^+^ T cells to maintain intestinal tolerance. In particular, ILC3s can act as antigen-presenting cells by expressing high levels of MHC class II within mesenteric lymph nodes. MHCII has been found to be highly expressed on RORγt+ ILCs that lack the expression of both T-bet and NKp46 while exhibiting the homogenous expression of CCR6 and the heterogeneous expression of CD4 [41]. However, different from classical antigen presentation, the interaction between CD4^+^ T cells and MHCII+ ILC3s results in the suppression of effector CD4^+^ T cells rather than their proliferation [41]. Indeed, intestinal ILC3s lack the expression of canonical co-stimulatory molecules (CD40, CD80, CD86) and, different from circulating ILC3s, remain absent even upon stimulation by IL-1β [42]. Moreover, ILC3s express CD25 molecules (conferring a high affinity to IL-2R), by which they can compete with T lymphocytes to bind IL-2, which is essential for their survival. In line with these observations, mice lacking MHC-expressing ILC3s exhibit increased frequencies of effector CD4^+^ T cells against commensal microbiota, leading to spontaneous intestinal inflammation [43]. To maintain immune tolerance against microbiota, antigen-presenting ILC3s are also able to induce microbiota-specific T regulatory (Treg) cells [44]. To ensure Treg differentiation, the migration of ILC3s to mesenteric lymph nodes via CCR7 is necessary, as well as their expression of integrin, aVß3 through which they promote the activation of latent TGF-β, which is required for Treg differentiation [45]. Interestingly, ILC3s produce large amounts of IL-2 in the gut, which is known to support Treg survival [46]. In addition, ILC3-derived GM-CSF promotes Treg cell differentiation by acting on mononuclear phagocytes [29]. ILC3s lacking one of these factors fail to maintain tolerance and lead to the expansion of pro-inflammatory Th17 cells against commensal bacteria [47]. ILC3s can also present antigens to T follicular helper (Tfh) cells by modulating B cell responses in germinal centers [48]. In the intestinal tract, the majority of activated B cells differentiate into IgAs producing cells, which is critical for mediating host–microbiota mutualism. IgAs promote physical segregation and shape the composition of microbiota by regulating bacteria colonization [49]. The dysregulation of ILC3s results in altered class-switched B cells and an increase in the high-affinity IgA coating of potential colitogenic bacteria. In addition, ILC3s regulate the production of IgA through the release of LTα and membrane-bound LTβ [40]. LTα sustains T cell-dependent IgA induction by favoring T cell homing to the gut, whereas LTβ interacts with DCs, upregulating their expression of inducible nitric oxide synthase (iNOS), which is necessary for IgA induction [40]. Finally, ILC3s can directly support IgA production by releasing B cell survival factors, such as the B cell-activating factor (BAFF) and A proliferation-inducing ligand (APRIL) [50].

Collectively, these data demonstrate that ILC3s are critical regulators of mucosal immune responses to microbiota (Figure 2), thus act as checkpoints to maintain host–microbiota homeostasis.

## 3. ILC3s, Microbiota and Colon Cancer

An important connection between colon cancer and the microbiota is the loss of immune tolerance to the latter, causing the development of an immune response that promotes chronic inflammatory diseases and increases the risk of colorectal cancer. An increasing number of studies have pointed toward the dysregulated role of ILC3s in modulating responses to the gut microbiome. The ability of MHCII-expressing ILC3s to establish immune tolerance to the microbiota is impaired during gut inflammation. In particular, MHCII expression is significantly reduced on ILC3s in pediatric Crohn’s disease patients compared to healthy tissues and this reduction results in an increase in the frequency of inflammatory T helper 17 (Th17) cells and the circulation of commensal bacteria-specific immunoglobulin G (IgG) titers [43]. The expansion of inflammatory Th17 has been associated with colonization with SFB [51]. Reductions in both the number and activity of ILC3s are associated with the overgrowth of SFB. Additionally, in mice lacking ILC3-specific MHCII, spontaneous intestinal inflammation occurs and the frequency of RORγt+ Treg cells is significantly decreased [43]. Accordingly, ILC3s are reduced in inflamed human guts and their frequency correlates with that of RORγt+ Treg cells, further supporting the concept that interactions between ILC3s and RORγt+ Treg cells are impaired during gut inflammation. Moreover, it has been reported that IBD-associated microbiota stimulate the production of TL1A by mononuclear phagocytes (MNPs) [31]. MNP-derived TL1A in turn increases the expression of the costimulatory molecule OX40 ligand (OX40L) on colonic ILC3s, leading to the proliferation and activation of pathogenic T cells [32]. Taken together, these data indicate that alterations in MHCII+ ILC3s caused by interrupting immune tolerance to commensal bacteria trigger pro-inflammatory pathways, which are important drivers of carcinogenesis. In agreement with a model of inflammation-associated cancer, ILC3s are diminished in human CRC and the deletion of MHCII on ILC3 is associated with CRC development and progression [52]. Thus, dysregulated dialog between MHCII+ ILC3s and T cells can alter microbiota colonization, which subsequently affects type-1 immunity in tumor microenvironments and promotes CRC progression. Interestingly, MHCII+ ILC3s are localized primarily within tertiary lymphoid structures (TLS) within CRC, which correlates with prolonged patient survival. In line with this finding, reductions in NKp44+ ILC3s in advanced-stage colon tumors are accompanied by a decrease in TLS density and the expression of lymphoid structure formation-related genes, such as LTα, LTβ and TNF, in these cells [53]. Remarkably, it has recently been reported that the microbiota can support the induction of TLS. Overacre-Delgoffe et al. showed that colonization with *Helicobacter hepaticus* (*Hhep*) induced *Hhep*-specific CD4^+^ T follicular helper cell expansion, which is crucial for driving TLS formation and promoting antitumor immunity [54]. While *Hhep* colonization induces a response dominated by Treg cells and Tfh cells during homeostasis [55], the expansion of inflammatory *Hhep*-specific Th1 and Th17 cells has been observed in experimental models of colitis. This is noteworthy since these models have highlighted the possible detrimental, rather than protective, role of ILC3s in colon cancer development. Indeed, during *Hhep*-driven cancers, ILC3s have also been suggested to play a pro-tumorigenic function since pathogenic double-negative Nkp46^−^ CD4^−^ ILC3s increase and their IL-22 production can promote tumorigenesis by supporting epithelial cell proliferation through Stat3 phosphorylation [56]. Along the same line, in another report, increases in commensal *C. albicans* in the gut induced IL-22 production by RORγt+ ILC3s, which is associated with CRC development [57]. The observation that IL-22 is related to the development of human colon cancer could be explained by the hypothesis that the same pathways that are critically involved in wound healing processes, once dysregulated, may favor tumorigenesis. In agreement with the deleterious effect of IL-22 in CRC, the sensing of microbial ligands by DCs downregulates IL-22BP expression, an antagonist of IL-22, and this downregulation increases inflammation and tumorigenesis in colitis-associated colon cancer models [58]. To summarize, the dysregulation of interactions between ILC3s and microbiota can significantly increase gut inflammation and promote CRC development.

## 4. ILC3s, Microbiota and CRC Immunotherapy

### 4.1. Colon Cancer and Response to ICI Therapy

ICIs have been shown to be effective in patients with metastatic colorectal cancer (mCRC) characterized by microsatellite instability (MSI-H) and/or mismatch repair deficiency (dMMR). Approximately 5% of patients with mCRC have tumors with an MSI-H/dMMR signature [59]. Mutations in mismatch repair genes can determine neoantigen burden, making these tumors highly immunogenic and responsive to immune checkpoint inhibitors [60,61,62]. Food and Drug Administration approvals of immunotherapeutic drugs for mCRC are the result of different clinical trials, such as KEYNOTE-164, KEYNOTE-177 and CheckMate-142. The first two explored the functionality and safety of pembrolizumab (anti-PD-1), while the third examined nivolumab (anti-PD-1) plus ipilimumab (anti-CTLA4) [63,64,65,66,67]. KEYNOTE-164 was an international phase II multicenter study that enrolled a total of 124 patients with locally advanced or metastatic MSI-H and/or dMMR CRC. With a median follow-up of 30 months, monotherapy with anti-PD-1 agents showed an objective response rate (ORR) of around 32% and a 2-year overall survival (OS) rate of 95% [63,64]. Similar results were observed in the phase 3 clinical trial KEYNOTE-77, which enrolled 153 patients. Monotherapy with pembrolizumab led to significantly longer progression-free survival when received as first-line therapy, with an overall response (complete or partial response) observed in 43.8% of patients and a 2-year OS rate of 66% [65]. In the phase II CheckMate-142 study, combination therapy with nivolumab plus ipilimumab as first-line treatment demonstrated durable clinical benefits and safety in all enrolled patients, with an ORR of 69% after a median of 29 months. Interestingly, combination therapy showed ORR benefits, regardless of BRAF or KRAS mutations [66,67]. Besides MSI-H advanced CRC, several ongoing trials are aiming to investigate the use of immunotherapy combinations in the advanced and early stages of both MSI-H and MSS CRCs.

### 4.2. Influence of ILC3–Microbiota Interactions on the Outcomes of ICI Therapy

Growing evidence has reported that the microbiota plays a crucial role in modulating ICI efficacy [68,69]. It has been reported that a “favorable” gut microbiome composition could enhance therapeutic response to ICIs, whereas antibiotic-treated patients and germ-free mouse models have shown a compromised efficacy of ICI therapy [68,69,70,71,72,73].

In a mouse model of colon cancer, the antitumor effect of CTLA-4 blockade was enhanced by specific *Bacteroides* species, which are associated with Th1-specific responses mediated by IL-12-producing intratumoral DCs [74]. Similarly, *A. muciniphila* restores PD1 blockade responsiveness by increasing the recruitment of IFN-γ+ CD4^+^ T cells within tumor microenvironments [70]. Along the same line, the antitumor efficacy of both anti-CTLA4 and anti-PD1 treatments is supported by a specific mixture of bacteria strains that are able to increase the frequency of tumor-infiltrating GrB+IFNγ+CD8^+^ T cells [68], which in turn suppress tumor growth and prolong progression-free survival in mice.

Recently, in a CRC mouse model, Goc et al. [52] demonstrated that the disruption of the dialog between MHCII+ ILC3s and T cells results in a microbiota compositional shift that favors immunotherapy resistance. This microbiota compositional shift concerns an unbalanced ratio between *Bacteroidales* and *Clostridiales*, which is associated with positive and negative ICI outcomes, respectively. In this regard, it has been reported that a decrease in either the number or activity of ILC3s results in marked reductions in commensal *Bacteroides* and *Lactobacilli* [40], strains that are positively associated with ICI responsiveness in CRC [75]. Importantly, alterations in ILC3/T cell crosstalk and immunotherapy resistance have already been associated within the context of chronic intestinal inflammation, such as inflammatory bowel diseases (IBDs), suggesting that the impairment of ILC3s could act in the very early stages of CRC development. ILC3–T cell interactions occur within TLS [52], which, in the context of ICI therapy, are associated with the regression of neoplastic lesions [76], longer disease-free survival and overall survival [77,78]. The density of TLS in CRC correlates with the frequency of ILC3s and with their expression of TLS formation-related genes [53]. Tumor-infiltrating ILC3s can support lymphoid neogenesis [18], promote vasculature changes [79,80] and favor the recruitment of leukocytes within tumors [18,19,20,21,22,23,24,25,26,27,28,29,30,31,32,33,34,35,36,37,38,39,40,41,42,43,44,45,46,47,48,49,50,51,52,53,54,55,56,57,58,59,60,61,62,63,64,65,66,67,68,69,70,71,72,73,74,75,76,77,78,79,80,81], suggesting that ILC3s could sustain TLS formation and affect therapeutic responses to ICIs. Moreover, within tumors, ILC3s are able to produce CXCL10 and recruit both CD4^+^ and CD8^+^ T lymphocytes, thereby promoting antitumor immune responses and enhancing the efficacy of checkpoint inhibition [82].

As a whole, microbiota manipulation represents an interesting therapeutic option for overcoming immunotherapy resistance. Fecal microbiota transplantation (FMT) from patients who respond to immunotherapy into non-responders or refractory patients restores ICI efficacy through Th1-dominant antitumor immune responses [70,71,72,73,74,75,76,77,78,79,80,81,82,83,84]. Analysis of the fecal bacterial content of melanoma patients who responded to anti-PD1 treatment [73] revealed the relative abundance of *Bifidobacterium*. Interestingly, it has been reported that *Bifidobacterium species* can produce aromatic lactic acids, including indolelactic acid (ILA), which can engage AhR, highly expressed by ILC3s, to mediate their immune responses. Short-chain fatty acid (SCFA) administration or diet integration with probiotics [85] or fiber as pectin [86], which enhances SCFA-derived microbiota production, can be used as interventional measures to promote ICI efficacy. SCFA administration combined with PD-1 therapy is associated with IFN-γ response [87], higher frequencies of CD4^+^ T cells [88] and CD8^+^ T cells [89] and higher progression-free survival [90]. SCFAs can have a direct effect on ILC3s expansion and functions in a GPCR-dependent manner [25,91] and it has been reported that the administration of SCFA-ameliorated anti-PD1/PD-L1 treatment reduces IL-17A-producing ILC3s [92]. Although immunological information is still limited in this model, including IL-22 production by ILC3s, these findings indicate the potential involvement of ILC3s in the mechanisms underlying the efficacy of SCFA–ICI combined therapy.

Although growing evidence has suggested that ILC3s may be critically involved in the outcome of ICI treatment influenced by the microbiota (Figure 3), further studies are required to clarify the role of ILC3s in the mechanisms underlying microbiota influence, aiming to improve both antitumor immune responses and immunotherapy efficacy in CRC.

## 5. Conclusions and Perspectives

Growing evidence has revealed that manipulating the microbiota could represent a potential adjunct to current anticancer immunotherapeutic strategies. However, the role of the human gut microbiota in ICI response is more intricate than previously considered. ICI responsiveness does not rely on the different bacteria strains that are simply present or absent in responders and non-responders but more likely depends on their complex interactions with the host immune system. Given their intimate relationship with the microbiota and their interplay with T cells in shaping microbiome composition, ILC3s may be in prime position to orchestrate antitumor immune responses in CRC and improve ICI efficacy.

Besides manipulating the microbiota, it may be intriguing to investigate whether supporting ILC3 functions could be a target for improving ICI efficacy. Different options could be taken into consideration, including targeting cytokines or the critical regulators of the tissue-restorative functions of ILC3s. In this regard, AhR is an attractive target since it has been shown to be essential for the maintenance of ILC3s in the intestines and the repression of inflammation [93]. Among cytokines, IL-1β is a known activator of ILC3s. ILC3s grown in the presence of IL-1β proliferate, produce IL-22 and exhibit high levels of RORγt [94,95].

Interestingly, it has been recently reported that the same IL-1β could increase CXCL10 production by ILC3s, be able to recruit both CD4^+^ and CD8^+^ T cells within tumors and improve anticancer immunity. Thus, IL-1β could affect multiple aspects of ILC3 biology, including antigen presentation capability [96]. Indeed, IL-1β drives the expression of HLA-DR and co-stimulatory molecules on peripheral blood ILCs. This effect is strongly inhibited by the anti-inflammatory cytokines TGF-β and IL-23, with potential implications for antitumor immunity and inflammation [95]. With this in mind, the rational design of ILC3-based therapeutic approaches should consider several factors and could represent the key for controlling immune responses to the microbiota and boosting antitumor immunity in CRC patients.

Finally, although targeting specific ILC3 functions may present potential novel approaches to combine with current ICI therapies, further experimental studies should be performed to confirm this promising therapeutic strategy.

## Figures and Tables

**Figure 1 cancers-15-02893-f001:**
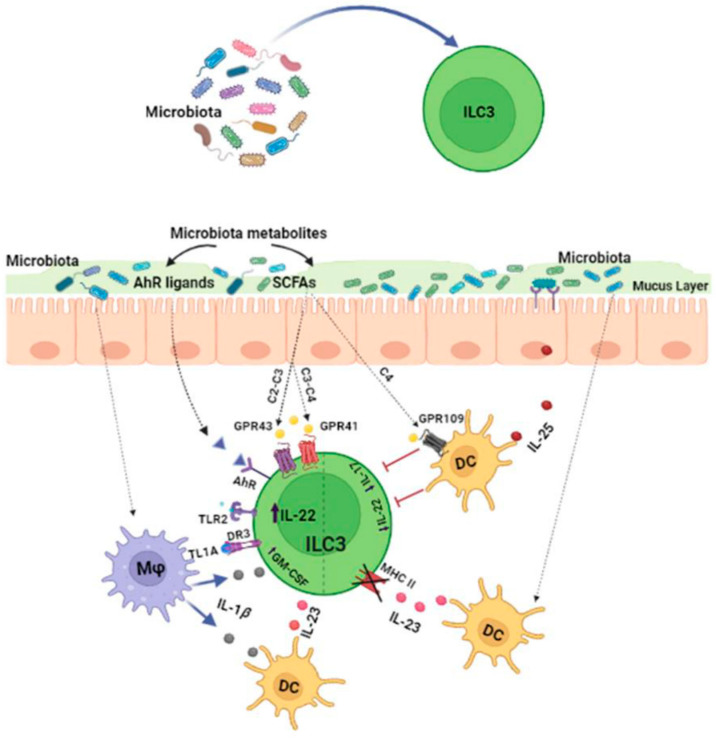
The microbiota regulates ILC3 functions through various mechanisms. The microbiota and its metabolites can directly affect ILC3 activity through a diverse array of receptors expressed on their surfaces. Alternatively, microbes can modulate ILC3 functions by stimulating the release of cytokines by myeloid or epithelial cells.

**Figure 2 cancers-15-02893-f002:**
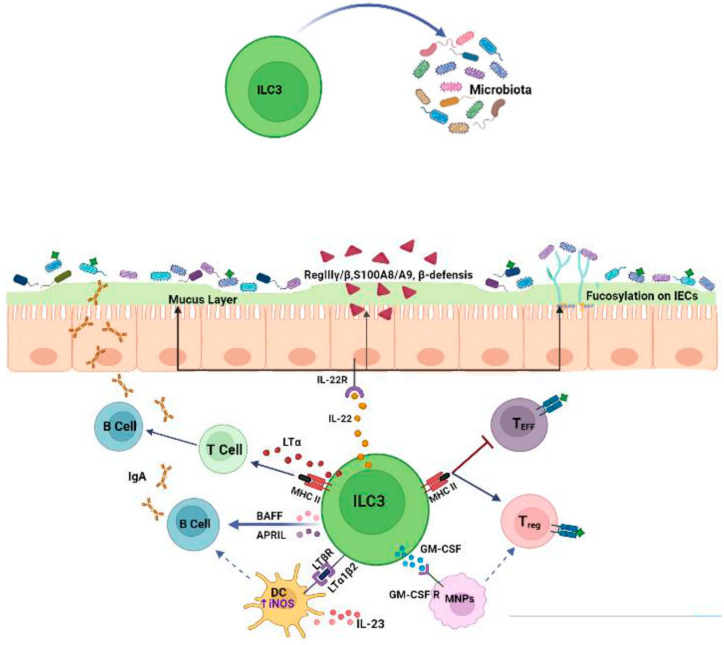
ILC3s regulate immune responses to commensal microbiota. ILC3s ensure the anatomical containment of bacteria, shape microbiota composition and orchestrate various tolerance mechanisms by acting on epithelial cells and adaptive immune populations.

**Figure 3 cancers-15-02893-f003:**
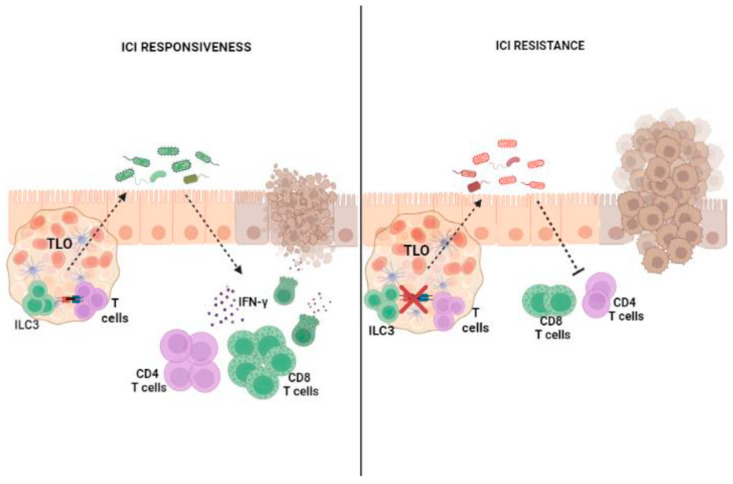
The role of ILC3s and the microbiota in responsiveness to ICIs. In CRC, dialog between ILC3s and T cells within the tertiary lymphoid organ (TLO) can support a microbiota composition that promotes antitumor immune responses and successful immunotherapy. Conversely, the dysregulation of this dialog promotes a microbiota that drives CRC progression and immunotherapy resistance.

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
