# Peer review of "Crosstalk between ILC3s and Microbiota: Implications for Colon Cancer Development and Treatment with Immune Check Point Inhibitors"

_cancers, 2023, doi:10.3390/cancers15112893_

Round 1

Reviewer 1 Report

This manuscript is a review of ILC3s and microbiota in colon cancer.  While overall it reads fairly well, there are numerous places where clarity is reduced by the phrasing.  Of most importance, Figure 1 is shown backwards, and needs to be reversed.

1.       (Figure 1) The words are given backwards, so the figure needs to be reversed.

2.       (line 76) What about TLR type 8?

3.       (line 282) Change “mice model” to “mouse model”.

4.       (line 283) Change “which associated” to “which is associated”.

5.       (line 292) Change “favour” to “favoured”.

6.       (line 293) Change “that are associated” to “that was associated”.

7.       (line 302) Change “longer disease-free survival and overall survival” to “longer disease-free and overall survival times”.

8.       (line 312) Change “not responders” to “non-responders”.

9.       (line 318) Change “pectine” to “pectin”.

10.   (line 322) Change “progression free” to “progression-free”.

11.   (line 342) Change “manipulating microbiota” to “manipulation of the microbiota”.

12.   (line 347) The phrase “Given their intimate relationship with microbiota” needs to be re-worked, as “their” refers to “human gut microbiota”, and so should have a relationship with microbiota.

13.   (line 356) A semi-colon should be used rather than a colon.

14.   “Summary” and “conclusion” are used several times (lines 138, 261, 328, and 356).  The earlier “summary” and conclusion” should be contextualized better, e.g. “To summarize …”.

Most of my comments above are on English language issues.  I recommend that a read over by a native English speaker be done.

Author Response

Please find our replies in bold.

This manuscript is a review of ILC3s and microbiota in colon cancer. While overall it reads fairly well, there are numerous places where clarity is reduced by the phrasing. Of most importance, Figure 1 is shown backwards, and needs to be reversed.

  1. (Figure 1) The words are given backwards, so the figure needs to be reversed.

That was somehow something that occured at editorial level, we originally submitted the right figure.

  1. (line 76) What about TLR type 8?

Concerning TLR8, information about its expression on ILC3 at both transcript and protein are completely lacking in literature.

  1. (line 282) Change “mice model” to “mouse model”.

Done

  1. (line 283) Change “which associated” to “which is associated”.

We actually do not find necessary this change

  1. (line 292) Change “favour” to “favoured”.

Done

  1. (line 293) Change “that are associated” to “that was associated”.

Again, not necessary because it is not referred to that single study

  1. (line 302) Change “longer disease-free survival and overall survival” to “longer disease-free and overall survival times”.

We would prefer to leave it as it is for the sake of clarity.

  1. (line 312) Change “not responders” to “non-responders”.

Done

  1. (line 318) Change “pectine” to “pectin”.

Done

  1. (line 322) Change “progression free” to “progression-free”.

Done

  1. (line 342) Change “manipulating microbiota” to “manipulation of the microbiota”.

Not necessary

  1. (line 347) The phrase “Given their intimate relationship with microbiota” needs to be re-worked, as “their” refers to “human gut microbiota”, and so should have a relationship with microbiota.

“their” actually refers to ILC3s.

  1. (line 356) A semi-colon should be used rather than a colon.

Done

  1. “Summary” and “conclusion” are used several times (lines 138, 261, 328, and 356). The earlier “summary” and conclusion” should be contextualized better, e.g. “To summarize …”.

Done

Comments on the Quality of English Language

Most of my comments above are on English language issues. I recommend that a read over by a native English speaker be done.

I actually do not really envisage major language issues, even according the reviewer’s comments.

Reviewer 2 Report

This manuscript reviewed studies of ILC3s and microbiota, illustrated crosstalk and bidirectional effects between them. Moreover, the authors described relationships of ILC3s, microbiota and diseases including colon cancer and CRC immunotherapy. This review article will be a good reference for connecting microbiota and immunome responses, and for providing patenting diseases treatment strategies targeting microbiota and ILC3s.

My suggestions are:

1.       in the whole manuscripts especially the cross talk between ILC3s and microbiota, microbiota is described as a general concept. Microbiota is very complicated, and different structures and conditions may have different effects on ILC3s. Vice versa, ILC3s will also affect the structures of microbiota. Therefore, it would be better to include more details/literatures about which kinds of microbiota relate to what effects on ILC3s, and how ILCs affect the structure of microbiota. This will also give a clearer perspective to develop promising therapeutic strategies.

2.        Please check the articles carefully. Labels in Figure 1 are reversed.

Author Response

Please find our replies in bold.

This manuscript reviewed studies of ILC3s and microbiota, illustrated crosstalk and bidirectional

My suggestions are:

  1. in the whole manuscripts especially the cross talk between ILC3s and microbiota, microbiota is described as a general concept. Microbiota is very complicated, and different structures and conditions may have different effects on ILC3s. Vice versa, ILC3s will also affect the structures of microbiota. Therefore, it would be better to include more details/literatures about which kinds of microbiota relate to what effects on ILC3s, and how ILCs affect the structure of microbiota. This will also give a clearer perspective to develop promising therapeutic strategies.

We understand the reviewer’concern and we agree with reviewer about the relevance of this issue. Unfortunately, only few studies have already investigated how gut microbial species might change in ILC3 deficient mice or, vice versa, the effect of specific microbial strains on ILC3 functions. However, we have now added and discussed these studies in the revised version of the manuscript.  

Please check the articles carefully. Labels in Figure 1 are reversed.

That was somehow something that occured at editorial level, we originally submitted the right figure. However, the figure has been now reloaded

Reviewer 3 Report

The review by Drommi et al. is centered on the microbiota-ILC3 interaction and how its alteration may contribute to colon cancer development and alteration on the therapeutic responce to immune checkpoint inhibitors. Overall the manuscript is well organized and contains updated literature revision on a relevant subject. I have just some minor suggestions to be considered before acceptance:

1-In section 3, Microbiota, ILC3 and colon cancer (lines 213 to 262) two different roles of ILC3 cells regarding colon carcinogenesis and progression are described. The tumor-preventing activity of ILC3 relaying on anti-inflamatory capacity of MHCII expresing ILC3 is clearly stated. On the other hand tumor promoting activity reports related to the ILC3-derived IL22 dependent progression of CCR is also mentioned (lines 250-261). These two different actions should be better separated and described as two opposite effects of ILC3 on the CCR landscape.

2. Minor corrections:  line 116: niacine is not a butyrate agonist but a GPR109a agonist

- Figure 1: the image seems to be inverted

- Correct duplications in literature numbering (References section, lines 370-580)

Author Response

Please find our replies in bold.

1-In section 3, Microbiota, ILC3 and colon cancer (lines 213 to 262) two different roles of ILC3 cells regarding colon carcinogenesis and progression are described. The tumor-preventing activity of ILC3 relaying on anti-inflamatory capacity of MHCII expresing ILC3 is clearly stated. On the other hand tumor promoting activity reports related to the ILC3-derived IL22 dependent progression of CCR is also mentioned (lines 250-261). These two different actions should be better separated and described as two opposite effects of ILC3 on the CCR landscape.

We appreciated the criticism. Section 3 has been amended according to reviewer’s suggestion.

  1. Minor corrections:  line 116: niacine is not a butyrate agonist but a GPR109a agonist

Thank you, that has been amended.

- Figure 1: the image seems to be inverted

That was an editorial problem. A new figure has been loaded.

- Correct duplications in literature numbering (References section, lines 370-580)

Again, that was created at editorial level by creating the draft according to MDPI template. Anyhow, we now corrected that.

Reviewer 4 Report

Title: Cross-talk between ILC3s and Microbiota: Implications for Colon Cancer Development and Treatment with Immune Checkpoint Inhibitors

Review Report:

In this article, the authors provide a comprehensive review of the functional interactions between Type 3 Innate Lymphoid Cells (ILC3s) and the microbiota in the context of homeostasis, gut inflammation, colorectal cancer (CRC) development, and response to immune checkpoint inhibitor (ICI) therapy. The authors build a strong case for the importance of understanding the cross-talk between ILC3s and the gut microbiota, given their critical roles in maintaining host-microbial mutualism and their potential implications in CRC progression and response to ICI therapy.

Strengths:

The authors have effectively provided a clear and concise summary of the current knowledge on the interactions between ILC3s and the gut microbiota, as well as their influence on the development of the gastrointestinal-associated lymphoid tissue (GALT), intra-epithelial lymphocytes (IELs), and IgA antibodies.

The review thoroughly discusses the role of ILC3s in the maintenance of the intestinal immune response to microbiota, particularly their involvement in various tolerance mechanisms, including the development of lymphoid structures and the reinforcement of intestinal epithelial integrity.

The authors present a well-organized and insightful overview of how dysregulation of ILC3s, resulting from environmental perturbations, may lead to impaired host-microbial mutualism, aberrant immune activation, and associated intestinal inflammation.

The review effectively highlights the potential implications of the ILC3-microbiota cross-talk in CRC development, progression, and response to novel immunotherapies, such as ICI therapy, thereby emphasizing the importance of further research in this area.

Weaknesses:

The review might benefit from a more in-depth discussion of the molecular mechanisms that underlie the interactions between ILC3s and microbiota, as well as their potential role in CRC development and ICI therapy responsiveness.

Please add more background information about immune checkpoint. (please cite: 1. Emerging phagocytosis checkpoints in cancer immunotherapy. Signal Transduct Target Ther. 2023 Mar 7;8(1):104. doi: 10.1038/s41392-023-01365-z. PMID: 36882399; PMCID: PMC9990587.  2. An Overview: The Diversified Role of Mitochondria in Cancer Metabolism. Int J Biol Sci. 2023 Jan 16;19(3):897-915. doi: 10.7150/ijbs.81609. PMID: 36778129; PMCID: PMC9910000.)

The authors could consider providing more information on the different subsets of ILC3s, such as LTI cells and IL-22-producing ILC3 populations, and their specific roles in maintaining host-microbial mutualism, gut inflammation, and CRC progression.

The review might benefit from a more detailed discussion of the current and potential future strategies to modulate ILC3-microbiota interactions for the prevention and treatment of CRC, as well as improving the response to ICI therapy.

Overall, this article is an informative and well-structured review that provides valuable insights into the cross-talk between ILC3s and the gut microbiota and their potential implications in CRC development and response to ICI therapy. The authors have demonstrated a deep understanding of the subject matter and have presented a strong case for the importance of further research in this area. With some improvements in the depth and breadth of the discussion, particularly concerning molecular mechanisms and therapeutic strategies, this review could be an excellent resource for researchers in the field.

Author Response

The review might benefit from a more in-depth discussion of the molecular mechanisms that underlie the interactions between ILC3s and microbiota, as well as their potential role in CRC development and ICI therapy responsiveness.

The manuscript has been now edited according to the reviewer’s suggestions.

Please add more background information about immune checkpoint. (please cite: 1. Emerging phagocytosis checkpoints in cancer immunotherapy. Signal Transduct Target Ther. 2023 Mar 7;8(1):104. doi: 10.1038/s41392-023-01365-z. PMID: 36882399; PMCID: PMC9990587.  2. An Overview: The Diversified Role of Mitochondria in Cancer Metabolism. Int J Biol Sci. 2023 Jan 16;19(3):897-915. doi: 10.7150/ijbs.81609. PMID: 36778129; PMCID: PMC9910000.)

Unfortunately the two suggested references, both surprisingly enough coming from the same group, do not seem to fit with the topic of our present review. We feel like rebutting this criticism.

The authors could consider providing more information on the different subsets of ILC3s, such as LTI cells and IL-22-producing ILC3 populations, and their specific roles in maintaining host-microbial mutualism, gut inflammation, and CRC progression.

We agree with reviewer’s criticism and, in the revised version of manuscript, we provided more detailed information on ILC3 subsets in both mice and humans. In particular, while in mice LTi cells have been clearly characterized, in human, it is still unclear how to distinguish LTi cells from the NCR+ ILC3s. Similarly, effector functions are also somewhat different between human and mouse ILC3 subsets. While both ILC3 subsets in mice are capable of producing IL-22, in human the IL-22 production is mainly confined to the NKp44+ populations. Thus, the information reported in literature about the specific role of each subset of ILC3 in maintaining host-microbial mutualism, gut inflammation, and CRC progression depends on the type of study conducted (mouse or human study), and the effector function of ILC3 involved (e.g. IL-22 production, antigen presentation capability). However, when specified in literature, we now reported details about the ILC3 subsets involved in the different biological processes described.

The review might benefit from a more detailed discussion of the current and potential future strategies to modulate ILC3-microbiota interactions for the prevention and treatment of CRC, as well as improving the response to ICI therapy.

According to reviewer’s suggestions, in the new version of manuscript, we extend our conclusions to better discuss potential ILC3-based therapeutic strategies able to modulate immune response towards microbioma and to boost anti-tumor immunity.

Reviewer 5 Report

This is a well-written manuscript and I thoroughly enjoyed reviewing this manuscript that only needs to undergo a few minor changes.

To improve the readability of the paper, the authors suggested to dividing big paragraphs.

The authors develop a unique table framework for ILC3s, microbiota and CRC immunotherapy, and I believe that they should highlight their originality much more.

Figure 1 is difficult to read and should be corrected.

Figures need higher resolution.

Line 278 – 281 – author suggested to crosscheck the font.

Author suggested to write about details of clinical studies.

I had difficulties understanding the conclusion is too long and suggest that the authors reformulate and simplify it.”

Author Response

Overall, this article is an informative and well-structured review that provides valuable insights into the cross-talk between ILC3s and the gut microbiota and their potential implications in CRC development and response to ICI therapy. The authors have demonstrated a deep understanding of the subject matter and have presented a strong case for the importance of further research in this area. With some improvements in the depth and breadth of the discussion, particularly concerning molecular mechanisms and therapeutic strategies, this review could be an excellent resource for researchers in the field.

This is a well-written manuscript and I thoroughly enjoyed reviewing this manuscript that only needs to undergo a few minor changes.

We appreciate the reviewer’s positive evaluation on this article.

To improve the readability of the paper, the authors suggested to dividing big paragraphs.

The authors develop a unique table framework for ILC3s, microbiota and CRC immunotherapy, and I believe that they should highlight their originality much more.

According to reviewer’s suggestions, in the revised version of this manuscript, we have now divided paragraph 4 in two sub-sections: subparagraph 4.1 describes the clinical response of colon cancer patients to ICI treatment while section 4.2 focuses on how ILC3s/microbiota cross-talk could influence the efficacy of ICI treatment. By dividing the paragraph and adding new titles, according to reviewer’suggeston, we could emphasize the role of ILC3 in CRC immunotherapy.

Figure 1 is difficult to read and should be corrected.

We have now provided a new Figure 1.

Figures need higher resolution.

It seems that something went wrong during manscript transformation in MDPI template. However, according to reviewer’s suggestion, we checked figure’s resolution, and we have now provided a better version of all figures present in the manuscript.

Line 278 – 281 – author suggested to crosscheck the font.

We checked the lines and fixed them according to the manuscript’s font.

Author suggested to write about details of clinical studies.

According to reviewer’s suggestion, we have now provided more information about clinical studies described in the new paragraph 4.1

I had difficulties understanding the conclusion is too long and suggest that the authors reformulate and simplify it.”

According to reviewer’s suggestion, we simplified the phrasing of the conclusion section to improve its readability.

Round 2

Reviewer 4 Report

strongly suggest for publication